# Synergistic Activity of Temocillin and Fosfomycin Combination against KPC-Producing *Klebsiella pneumoniae* Clinical Isolates

**DOI:** 10.3390/antibiotics13060526

**Published:** 2024-06-04

**Authors:** Venera Costantino, Luigi Principe, Jai Mehat, Marina Busetti, Alessandra Piccirilli, Mariagrazia Perilli, Roberto Luzzati, Verena Zerbato, Antonietta Meliadò, Roberto La Ragione, Stefano Di Bella

**Affiliations:** 1Microbiology Unit, Trieste University Hospital (ASUGI), 34128 Trieste, Italy; venera.costantino@asugi.sanita.fvg.it (V.C.); marina.busetti@asugi.sanita.fvg.it (M.B.); 2Clinical Microbiology and Virology Unit, Great Metropolitan Hospital “Bianchi-Melacrino-Morelli”, 89128 Reggio di Calabria, Italy; luigi.principe@ospedalerc.it (L.P.); antonietta.meliado@ospedalerc.it (A.M.); 3Department of Microbial Sciences, School of Biosciences, University of Surrey, Guildford GU2 7XH, UK; jw.mehat@surrey.ac.uk (J.M.); r.laragione@surrey.ac.uk (R.L.R.); 4Department of Biotechnological and Applied Clinical Sciences, University of L’Aquila, 67100 L’Aquila, Italy; alessandra.piccirilli@univaq.it (A.P.); mariagrazia.perilli@univaq.it (M.P.); 5Clinical Department of Medical, Surgical and Health Sciences, Trieste University, 34129 Trieste, Italy; roberto.luzzati@asugi.sanita.fvg.it; 6Infectious Diseases Unit, Trieste University Hospital (ASUGI), 34125 Trieste, Italy; verena.zerbato@asugi.sanita.fvg.it; 7Department of Comparative Biomedical Sciences, School of Veterinary Medicine, University of Surrey, Guildford GU2 7XH, UK

**Keywords:** antibiotic, combination, fosfomycin, *Galleria mellonella*, in vivo, KPC-producing *Klebsiella pneumoniae*, resistance, synergism, temocillin, therapy

## Abstract

Infections caused by KPC-producing *K. pneumoniae* continue to pose a significant clinical challenge due to their emerging resistance to new antimicrobials. We investigated the association between two drugs whose roles have been repurposed against multidrug-resistant bacteria: fosfomycin and temocillin. Temocillin exhibits unusual stability against KPC enzymes, while fosfomycin acts as a potent “synergizer”. We conducted in vitro antimicrobial activity studies on 100 clinical isolates of KPC-producing *K. pneumoniae* using a combination of fosfomycin and temocillin. The results demonstrated synergistic activity in 91% of the isolates. Subsequently, we assessed the effect on *Galleria mellonella* larvae using five genetically different KPC-Kp isolates. The addition of fosfomycin to temocillin increased larvae survival from 73 to 97% (+Δ 32%; isolate 1), from 93 to 100% (+Δ 7%; isolate 2), from 63 to 86% (+Δ 36%; isolate 3), from 63 to 90% (+Δ 42%; isolate 4), and from 93 to 97% (+Δ 4%; isolate 10). Among the temocillin-resistant KPC-producing *K. pneumoniae* isolates (24 isolates), the addition of fosfomycin reduced temocillin MIC values below the resistance breakpoint in all isolates except one. Temocillin combined with fosfomycin emerges as a promising combination against KPC-producing *K. pneumoniae*, warranting further clinical evaluation.

## 1. Introduction

Infections caused by *Klebsiella pneumoniae* carbapenemase (KPC)-producing *Klebsiella pneumoniae* are associated with high mortality rates [1]. In Europe, from 2007 to 2015, the burden of carbapenem-resistant *K. pneumoniae* increased by 6.16 times in terms of both the number of infections and number of deaths [2]. In patients with bloodstream infections (BSI), the resistance to carbapenems is associated with excess mortality [3]. It has been demonstrated that initiating the appropriate antibiotic therapy as early as possible for KPC-producing *K. pneumoniae* infections reduces mortality [4]. Currently, apart from colistin and eravacycline, we have four new β-lactam drugs with activity against KPC-producing *K. pneumoniae*: cefiderocol, ceftazidime/avibactam, imipenem/relebactam and meropenem/vaborbactam. However, the microbial landscape is dynamic, with resistances emerging easily and increasingly reported [5]. According to the Infectious Diseases Society of America (IDSA), the preferred antibiotics for the treatment of infections outside of the urinary tract caused by carbapenem-resistant *Enterobacterales* (CRE) if KPC production is present are the following: meropenem–vaborbactam, followed by ceftazidime–avibactam, and then imipenem cilastatin–relebactam. Cefiderocol is listed as an alternative option [6]. The European Society of Clinical Microbiology and Infectious Diseases (ESCMID) guidelines recommend meropenem–vaborbactam or ceftazidime–avibactam for patients with severe CRE infections [7]. IDSA and the ESCMID does not recommend combination therapies for CRE infections, if susceptible to the new beta-lactams/beta-lactamase inhibitors. Moreover, antimicrobial stewardship principles advocate for the use of an antibiotic with the narrowest spectrum possible, at least for monomicrobial infections. In this context, temocillin, an old antibiotic, emerges as an interesting option. Temocillin is a semisynthetic derivative of ticarcillin, a carboxypenicillin commonly used in the 1980s and 1990s as an agent against Gram-negative bacteria [8]. Temocillin is used mainly for urinary tract infections (UTI), pneumonia, abdominal infections, and BSI. Generally, it is well tolerated and is associated with a low rate of *C. difficile* infections [8]. Temocillin exhibits unusual stability against extended-spectrum β-lactamases (ESBL), AmpC and KPC enzymes [8], raising interest in drug repurposing. Indeed, currently, there are two ongoing clinical trials whose aim is to compare between temocillin and carbapenems in complicated UTI [9] and BSI [10] due to Gram-negative bacteria resistant to third-generation cephalosporins. From an antimicrobial stewardship perspective, the inactivity of temocillin against bacteria producing metallo-beta-lactamase and OXA enzymes may be seen as an advantage, making it a strict anti-KPC agent. Fosfomycin is another old antibiotic with unique properties. It is a small molecule capable of synergizing with several other antibiotics, especially beta-lactams. It is a broad-spectrum antibiotic active against antimicrobial-resistant and multidrug-resistant (MDR) Gram-positive and Gram-negative bacteria, including methicillin-resistant *Staphylococcus aureus* (MRSA), vancomycin-resistant enterococci (VRE), *Enterobacterales*, and *Pseudomonas aeruginosa* [11]. Moreover, both fosfomycin and temocillin have proven to be stable for at least 24 h in elastomeric pumps, making them excellent options for outpatient parenteral antimicrobial therapy [12,13]. Temocillin offers the pharmacokinetic/pharmacodynamic advantages of a β-lactam drug, while fosfomycin provides the advantages of a small molecule, with good tissue penetration and anti-biofilm activity. The combination of the two molecules theoretically helps to reduce the bacterial minimum inhibitory concentration (MIC) against antibiotics without requiring a higher dose of the single drugs [14], potentially increasing bacterial killing and limiting drug-related toxicity. In the setting of MDR infections, the number of in vitro and in vivo studies on different antimicrobial combination treatments are increasing [11,15]. To the best of our knowledge, currently there are only a few studies on the synergistic activity in vitro and in vivo of temocillin and fosfomycin against *Enterobacterales*. Only one of these is focused on a KPC-producing *E. coli* infection in a murine model of peritonitis [16,17]. In light of this premise, we aimed to test the in vitro and in vivo potential of temocillin and fosfomycin combinations against clinical isolates of KPC-producing *K. pneumoniae*.

## 2. Results

### 2.1. Microbiological Findings

One hundred non-repetitive *K. pneumoniae* clinical isolates were investigated. All of the isolates were recovered from different patients (n = 100) and were identified at the species level via MALDI-TOF mass spectrometry. All *K. pneumoniae* were KPC producers resistant to carbapenems (MICs > 1 mg/L). The isolates were from rectal swabs (n = 62), urine (n = 29), blood culture (n = 7) and bronchial aspirate (n = 2). Isolates were recovered from patients hospitalized in the medical department (n = 68), surgical department (n = 9), intensive care unit (n = 22) and emergency department (n = 1). In particular, among the medical department patients, about 40% were from the infectious diseases unit (n = 27). Thirty-four isolates were resistant to temocillin (MICs > 16 mg/L), while no resistant isolate was recovered for fosfomycin (MICs ≤ 32 mg/L). The fosfomycin–temocillin combination showed a prevalent synergistic activity against 91% of KPC-producing *K. pneumoniae* isolates, while additive activity was obtained against 8% of them. Only in one case was indifferent activity revealed. No antagonism was detected (Table 1). In particular, synergistic activity lowered temocillin MIC values under the resistance breakpoint (MICs ≤ 16 mg/L) for all isolates except one (99%), with an FIC index ranging from 0.21 to 1 considering all combinations (Table 1).

### 2.2. In Vivo Findings

Temocillin was determined to not be toxic to *G. mellonella* larvae, either alone or in combination with fosfomycin. A variable degree of synergism was observed between temocillin and fosfomycin in time-kill assays of *K. pneumoniae*. This synergism was recapitulated in vivo using the *G. mellonella* infection model, using select *K. pneumoniae* isolates. We observed clear increases in survival rates of larvae treated with combination therapy compared to either agent alone (Figure 1).

Larvae infected with *K. pneumoniae* isolate 1 and treated with a combination of temocillin and fosfomycin exhibited a survival rate of 96.6% compared to 90% for fosfomycin alone, and 73.3% for temocillin alone (*p* = 0.0252, n = 30). In larvae infected with *K. pneumoniae* isolate 2, a combination of both antimicrobials resulted in a 100% survival rate, relative to an 86% survival rate for temocillin monotherapy. Fosfomycin monotherapy led to a 93% survival rate, with only a single larva succumbing to challenge at 72 h post-infection. *K. pneumoniae* isolate 3 elicited 90% mortality; however, the combination of temocillin at 24 and fosfomycin at 8 significantly improved the survival rate relative to 86% relative to temocillin alone 63% (*p* = 0.0768, n = 30), or fosfomycin alone 53% (*p* = 0.0205, n = 30). Combination treatment of larvae infected with *K. pneumoniae* isolate 4 resulted in a survival rate of 90% compared to a survival rate of 46% without antibiotic therapy, and 63% for temocillin alone (*p* = 0.0344, n = 30) and 73% for fosfomycin alone (*p* = 0.1915, n = 30). The survival rate of larvae infected with *K. pneumoniae* isolate 10 was 76.6%. Treatment of larvae with temocillin increased survival to 93.3%, and treatment with fosfomycin alone increased survival to 96.6%. Combination therapy with both of these agents resulted in a survival rate of 96.6% (Figure 1).

### 2.3. Molecular Characterization by WGS of 1, 2, 3, 4 and 10 K. pneumoniae Isolates

The whole genome sequence (WGS) of five *K. pneumoniae* clinical isolates was performed, and the antibiotic resistance genes (ARGs) and mobile genetic elements (MGEs) are reported in Table 2 and Table 3. The isolates 2 and 4 belonged to ST307, isolate 1 belonged to ST512 and isolate 3 belonged to ST101. The ST of isolate 10 is not present in the MLST database of the Institut Pasteur; for this reason, we have inserted it as an unknown sequence type. As shown in Table 2, all five clinical isolates harbored KPC-β-lactamases, specifically KPC-2 (isolates 2, 4 and 10) and KPC-3 (isolates 1 and 3). Moreover, other β-lactamases including CTX-M-14 (isolate 10), CTX-M-15 (isolates 2, 3 and 4), SHV-11 (isolate 1), SHV-28 (isolates 2, 3 and 4), OXA-1 (isolates 2, 3 and 4), OXA-10 (isolate 4) and TEM-1B (isolates 3 and 10) were found. In all five strains, several chromosomal mutations that contributed to increase the resistance against carbapenems and cephalosporins were identified in *ompK36* and *ompK37* (Table 2). The chromosomal mutations P161R, G164A, F172S, R173G, L195V, F197I and K201M were found in the *acrR* of isolates 1, 2 and 4. AcrR is the main regulator of the AcrAB multidrug efflux pump that shows the widest substrate specificity, including fluoroquinolones. The presence of *aac(6*′*)-Ib-cr* and *qnrB1* increased resistance to fluoquinolones in isolates 2, 3 and 4. In isolate 10, resistance to ciprofloxacin was mediated by mutations in *gyrA* (S83L and D87N) and *parC* (S80I). The *fosA5* and *fosA6* genes were detected in these strains with the exception of isolate 10. Other ARGs conferring resistance to macrolides, aminoglycosides, tetracyclines, trimethoprim, chloramphenicol and rifampicin were also detected. Concerning mobile genetic elements (MGEs), transposon Tn*4401*, harboring the *bla*_KPC-2_ and *bla*_KPC-3_ genes, was identified in all five clinical isolates. In addition, IncFIB(K), IncFII, IncFII(K), IncR, ColKP3, ColRNAI and IncC plasmids were also identified (Table 3).

## 3. Materials and Methods

Clinical bacterial isolates from various sample types (rectal swabs, urine, blood, bronchial aspirate) collected from January 2021 and August 2022 at the microbiology unit of the University Hospital of Trieste (Azienda Sanitaria Universitaria Giuliano Isontina) underwent screening for carbapenemase production using selective chromogenic agar medium (bioMérieux, Marcy-l’Etoile, France) as part of routine laboratory activities. Only one isolate per patient was studied. Species identification on bacterial isolates on selective medium was performed using MALDI-TOF mass spectrometry (bioMérieux, Marcy-l’Etoile, France), while antimicrobial susceptibility was assessed with the Vitek2 system (bioMérieux, Marcy-l’Etoile, France). The susceptibility profile of temocillin and fosfomycin was assessed using the Etest (bioMérieux, Marcy-l’Etoile, France). Isolates with meropenem MIC values greater than 0.125 mg/L underwent further investigation for carbapenemase production via PCR (Cepheid, Sunnyvale, CA, USA), according to the EUCAST criteria for detection of resistance mechanisms (https://www.eucast.org, accessed on 1 March 2024). Synergy tests between temocillin and fosfomycin were conducted for KPC-producing *K. pneumoniae* isolates. Although the recommended testing for fosfomycin is agar dilution, we used the Etest also for fosfomycin since it allowed us to perform synergy testing for the standard MIC of drugs A and B before creating the synergy. Synergy testing methods have been used to assess the interactions of antibiotic combinations in vitro. The goal of synergy testing is to assess the in vitro interactions of antimicrobial combinations, in order to determine whether the effect of the two antimicrobials is greater than the sum of their individual activities. Conventional, single-drug testing with the MIC test strip (gradient diffusion method) relies on the diffusion of a continuous concentration gradient of antimicrobial from an impregnated strip into solid agar. MIC test strips are placed on agar medium that has been inoculated with a lawn of the test organism. The “MTS Synergy Applicator System” was used for the synergy test. In particular, the MIC test strip (MTS) of the antibiotic (A) was positioned on the MTS synergy applicator platform. The MTS (antibiotic A) was adjusted such that the MIC value of the first antibiotic (MICA) was placed at the intersection of the base. An MTS of the second antibiotic (B) was placed on the second base. The second STD (antibiotic B) was adjusted such that the MIC value of the second antibiotic (MICB) was positioned at the base intersection and intersects MTS-antibiotic A at its MIC value. Finally, the plates were incubated according to the standard MTS procedure for the specific microorganism [18]. A four-fold reduction in the MIC values of the antibiotics in combinations, in comparison with MIC values alone, was considered synergistic (FIC index ≤ 0.5). An FIC index (FICI) between >0.5 and ≤1 was considered additive interaction; an FICI between >1 and ≤4 was considered indifferent interaction; an FICI > 4 was considered antagonistic. EUCAST breakpoints were used to value susceptibility to temocillin and fosfomycin [19].

### 3.1. In Vivo Toxicity of Antibiotic Compounds

We have previously determined that fosfomycin is not toxic to *Galleria mellonella* larvae at doses of up to 512 mg/L [20]. The in vivo toxicity of temocillin was determined according to the approach described previously for zidovudine and fosfomycin [21]. *G. mellonella* were purchased from LiveFoods UK Ltd. (Axbridge, UK) and stored at 15 °C prior to use. Only larvae weighing between 0.7 g and 1.3 g, showing no discoloration or injury, were used in the assays. Groups of 10 larvae were injected into the proleg using a 26S gauge sterile syringe with either temocillin (48 mg/L) and separately, a combination of temocillin (48 mg/L) and fosfomycin (32 mg/L). These concentrations are equivalent to the highest MIC observed for each agent against *K. pneumoniae* isolates used in this study. Larvae mock-infected with phosphate-buffered saline (PBS) were used as injection controls. Following inoculation, all larvae were incubated aerobically at 37 °C for 72 h and were assessed for mortality at 24 h intervals. Larvae were classed as dead when movement was not detected when applying stimulus. This assay was performed in triplicate.

### 3.2. In Vivo Challenge and Antibiotic Therapy

A single colony of the test *K. pneumoniae* isolates was harvested from a freshly streaked Luria-Bertani (LB) agar plate. Then, the colony was transferred to 10 mL of lysogeny broth and incubated at 37 °C, with gentle agitation (150 rpm) under aerobic conditions for 16 h. Following incubation, the cultures were serially diluted 10-fold in PBS and enumerated on LB agar in triplicate. Groups of 10 *G. mellonella* larvae were placed in Petri dishes, and challenged with 5000 CFU/mL of either *K. pneumoniae* isolate 1, 2, 3, 4 and 10—reference, using the top-left proleg (Table 1). Additionally, a group of 10 larvae were mock-infected with 10 µL of sterile PBS. Within 15 min of infection, a second injection into the top right proleg was performed to administer fosfomycin, temocillin, a combination of both agents or PBS control. The concentration of each drug was equal to that used in combination with time-kill assays. Larvae were incubated aerobically at 37 °C and scored for mortality at 0, 24, 48 and 72 h post-infection. All assays were performed in triplicate, and the data were plotted using GraphPad Prism 8.4.3 software (GraphPad Software Inc., San Diego, CA, USA).

### 3.3. Whole-Genome Sequencing and Bioinformatics Analysis of 1, 2, 3, 4 and 10 K. pneumonia Isolates

Short-read sequencing libraries were prepared with an Illumina DNA Prep Kit (Illumina Inc., San Diego, CA, USA) and sequenced on an Illumina MiSeq instrument with a 2 × 300 bp paired-end protocol, as previously described (REF-Salmonella). Quality control and sequences filtering were performed using DRAGEN FastQC + MultiQC v3.9.5 (https://basespace.illumina.com/apps/12821810/DRAGEN-FastQC-MultiQC?preferredversion, access date: 23 May 2024). Paired-end reads were assembled with SPAdes Genome Assembler v3.9.0. (https://basespace.illumina.com/apps/3047044/SPAdes-Genome-Assembler?preferredversion, access date: 23 May 2024). ResFinder 4.5.0 was used to detect acquired antimicrobial resistance genes (ARGs) (https://www.genomicepidemiology.org/services/, access date: 24 May 2024) and chromosomal mutations. Mobile genetic elements (plasmids, insertion sequences, transposons) were identified using MobileElement Finder v1.0.3 (https://cge.food.dtu.dk/services/MobileElementFinder/, access date: 24 May 2024). The genome was also assigned to ST using MLST 2.0.9 (https://cge.food.dtu.dk/services/MLST/, access date: 24 May 2024).

## 4. Discussion

KPC-producing isolates represent an important public health issue. Recently, emerging data on food-producing animals and their products, as reservoirs for KPC-producing isolates, are also being reported in different countries [22,23]. New beta-lactams/beta-lactamase inhibitors are now considered a first-line treatment option against KPC-producing isolates. Nevertheless, the widespread clinical use of ceftazidime/avibactam has forced CRE to mutate, in order to adapt to the increasing antibiotic pressure. Several isolates carrying different KPC variants and resistant to ceftazidime/avibactam are emerging worldwide [5]. Resistance to ceftazidime/avibactam in *Enterobacterales* is commonly due to three different mechanisms: enzymatic alterations causing inactivation of the antibiotics; modification of the antibiotic target or expressions of an alternative target; and changes in cell permeability or expression of efflux pumps. Modification of β-lactamase hydrolytic properties due to specific mutations within class A carbapenemase is the most common mechanism related to ceftazidime/avibactam resistance in *Enterobacterales* [24]. By March 2023, 145 *bla*_KPC_ variants were registered in the National Center for Biotechnology Information (NCBI) database, all derived from mutations of *bla*_KPC-2_ or *bla*_KPC-3_. Overall, *bla*_KPC-4_ (n = 26), *bla*_KPC-33_ (n = 9), *bla*_KPC-12_ (n = 8), *bla*_KPC-6_ (n = 5), *bla*_KPC-71_ (n = 4), *bla*_KPC-10_ (n = 3), *bla*_KPC-76_ (n = 3), *bla*_KPC-44_ (n = 3), *bla*_KPC-25_ (n = 2), *bla*_KPC-36_ (n = 2), *bla*_KPC-5_ (n = 2) and *bla*_KPC-90_ (n = 2) were mutants from *bla*_KPC-2_. These KPC-2 variants were mainly identified from the US (n = 35), China (n = 20) and Italy (n = 3). In contrast, *bla*_KPC-31_ (n = 8), *bla*_KPC-66_ (n = 4), *bla*_KPC-67_ (n = 3), *bla*_KPC-18_ (n = 2), *bla*_KPC-29_ (n = 2), *bla*_KPC-40_ (n = 2), *bla*_KPC-49_ (n = 2), *bla*_KPC-61_ (n = 2) and *bla*_KPC-70_ (n = 2) were mutants from *bla*_KPC-3_. These *bla*_KPC-3_ variants were mainly reported from Italy (n = 15), the US (n = 8) and France (n = 1) [25]. Resistance to ceftazidime/avibactam represents a serious cause for concern, with cases of resistance especially reported from the US, Greece and Italy. Moreover, an important rate of KPC producer strains worldwide (about 30%) shows a baseline resistance, mostly due to impermeability mechanisms. This is an important point, because isolates with baseline resistance to ceftazidime/avibactam could represent a reservoir of resistance that could be potentially enhanced under inappropriate ceftazidime/avibactam-based treatment. These isolates commonly remain susceptible to meropenem/vaborbactam, imipenem/relebactam and cefiderocol, among the new beta-lactam options. Moreover, some isolates carrying specific KPC variants or increased enzyme expression, in association with porin defects/loss, showed resistance also to meropenem/vaborbactam and imipenem/relebactam [25,26]. Although resistance to meropenem/vaborbactam has been associated with decreased expression of *ompK35* and *ompK36* and concomitantly increased expression of *bla*_KPC_, MIC seems to be unaffected by an increase in expression of the *bla*_KPC_ gene and efflux pump (*acrB*), or decreased expression of *ompK35* alone [27]. Among Enterobacterales and other Gram-negative bacteria efflux pump systems, in particular, AcrAB-TolC are common resistance mechanisms against multiple antibiotic classes. Overexpression of *acrAB* in association with inactivated OmpK35 and OmpK36 porins increased the MIC of meropenem/vaborbactam [28]. Previous studies demonstrated that imipenem/relebactam resistance was associated with mutations, resulting in a non-functional OmpK35 and OmpK36 porins in KPC-producing *K. pneumoniae* strains, or AmpC overexpression in combination with porins loss [29]. Similarly, resistance to cefiderocol, although uncommon, is increasingly being reported [30,31]. In this context, some old antibiotics such as temocillin and fosfomycin could represent beta-lactams/beta-lactamase inhibitors and cefiderocol-sparing options, but also may provide pharmacokinetic/pharmacodynamic and biochemical advantages (e.g., central nervous system penetration, anti-biofilm activity, etc.), hence improving the antimicrobial armamentarium against KPC-producing isolates. Fosfomycin, in combination with beta-lactams, represents a classical (and historical) synergistic option. Beta-lactam antibiotics target the bacterial cell wall’s penicillin-binding proteins, while fosfomycin inhibits the peptidoglycan precursor UDP N-acetylmuramic acid involved in peptidoglycan biosynthesis, hence providing a greater chance of synergistic/additive effects against Gram-negative isolates. Among the beta-lactams, the best synergistic activity against KPC-producing *K. pneumoniae* isolates was found in ceftazidime/avibactam. Fosfomycin demonstrated synergistic activity against KPC-producing *K. pneumoniae* isolates with tigecycline, colistin and aminoglycosides [11]. Moreover, recently, synergistic activity was also reported between fosfomycin and meropenem/vaborbactam [32,33], as well as between fosfomycin and cefiderocol [34]. In the present study, we assessed the antimicrobial interactions among these drugs against KPC-producing *K. pneumoniae* clinical isolates, demonstrating a significant in vitro and in vivo synergistic activity. In particular, in vitro synergistic activity between temocillin and fosfomycin was demonstrated against 91% of KPC-producing *K. pneumoniae* isolates, lowering the temocillin MIC values under the resistance breakpoint in all cases. In the animal study, we tested the temocillin and fosfomycin combination on five different KPC-producing *K. pneumoniae* isolates. The addition of fosfomycin to temocillin increased larvae survival from 73 to 97% (+Δ 32%; isolate 1), from 93 to 100% (+Δ 7%; isolate 2), from 63 to 86% (+Δ 36%; isolate 3), from 63 to 90% (+Δ 42%; isolate 4) and from 93 to 97% (+Δ 4%; isolate 10). To our knowledge, another study evaluated the in vitro and in vivo synergism of fosfomycin and temocillin against *Enterobacterales* KPC-producers, in particular an *E. coli* KPC-3 producer strain. In vitro, the addition of fosfomycin reduced the temocillin MIC 16-fold for this isolate of *E. coli* KPC-3, with an FICI of 0.562, showing an additive effect. This was eventually demonstrated in vivo in murine models of peritonitis, with a survival rate of 93% in mice infected with *E. coli* KPC-3 and treated with temocillin plus fosfomycin [16]. In our study, there were no strains resistant to fosfomycin. In cases of temocillin resistance, the co-administration of fosfomycin resulted in lowered temocillin MIC values under the resistance breakpoint (MICs ≤ 16 mg/L) for all isolates except one (99%). In our infected larvae, the combination therapy with temocillin and fosfomycin resulted in a survival rate of 96.6%, showing results similar to those already reported in the study of Berleur et al. [16]. The combination of fosfomycin and temocillin was also found to be synergistic in 27% of Gram-negative invasive isolates (*E. coli* or *K. pneumoniae* with a relatively low percentage of multiple drug-resistant organisms) of an observational study in the UK. However, no KPC-producing *Enterobacterales* were reported in this study [17]; thus, a comparison is not possible with our findings.

Our study has limitations. Firstly, it is monocentric. The epidemiology, especially for Gram-negative infections, could vary significantly among different areas within the same country. Thus, our study reflects only the local epidemiology. Secondly, we did not include many invasive clinical isolates.

In conclusion, this study represents the first in vitro and in vivo description of temocillin–fosfomycin synergistic interactions against KPC-producing *K. pneumoniae* isolates. Temocillin combined with fosfomycin emerges as a promising combination against KPC-producing *K. pneumoniae*, warranting further clinical investigations.

## Figures and Tables

**Figure 1 antibiotics-13-00526-f001:**
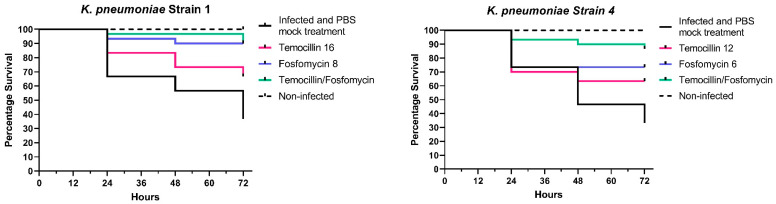
In vivo combination therapy (fosfomycin + temocillin)—survival curves.

**Table 1 antibiotics-13-00526-t001:** Fosfomycin + temocillin: in vitro synergy testing.

#	MIC FOS	MIC TEM	MIC FOS/TEM	MIC TEM/FOS	FICI	Interpretation
1	8	16	1	2	0.25	SYNERGISM
2	12	16	3	4	0.5	SYNERGISM
3	8	24	3	6	0.62	ADDITIVE
4	6	12	1.5	2	0.41	SYNERGISM
5	4	8	1	2	0.5	SYNERGISM
6	8	12	2	3	0.5	SYNERGISM
7	6	8	1	2	0.41	SYNERGISM
8	12	16	1	3	0.27	SYNERGISM
9	12	24	3	6	0.5	SYNERGISM
10	4	6	0.75	1.5	0.43	SYNERGISM
11	6	12	1.5	2	0.41	SYNERGISM
12	16	24	3	8	0.52	ADDITIVE
13	12	24	3	4	0.41	SYNERGISM
14	4	8	0.75	2	0.43	SYNERGISM
15	12	16	2	3	0.35	SYNERGISM
16	24	32	18	16	1.25	INDIFFERENT
17	16	32	4	6	0.43	SYNERGISM
18	16	24	4	6	0.5	SYNERGISM
19	12	16	2	4	0.41	SYNERGISM
20	8	16	2	3	0.43	SYNERGISM
21	8	16	1	2	0.25	SYNERGISM
22	12	16	1.5	3	0.31	SYNERGISM
23	4	6	0.5	1.5	0.37	SYNERGISM
24	6	8	1	1.5	0.35	SYNERGISM
25	8	12	2	3	0.5	SYNERGISM
26	32	48	6	12	0.43	SYNERGISM
27	12	16	3	4	0.5	SYNERGISM
28	8	16	1.5	4	0.43	SYNERGISM
29	12	24	2	6	0.41	SYNERGISM
30	24	32	6	8	0.5	SYNERGISM
31	16	32	4	8	0.5	SYNERGISM
32	8	12	2	3	0.5	SYNERGISM
33	4	6	0.5	1	0.29	SYNERGISM
34	6	8	0.75	2	0.37	SYNERGISM
35	8	12	1	3	0.37	SYNERGISM
36	16	24	3	6	0.43	SYNERGISM
37	24	48	6	12	0.5	SYNERGISM
38	8	16	2	6	0.62	ADDITIVE
39	12	16	1	3	0.27	SYNERGISM
40	4	6	1	1.5	0.5	SYNERGISM
41	12	16	2	3	0.35	SYNERGISM
42	8	16	0.75	3	0.28	SYNERGISM
43	12	16	1	2	0.20	SYNERGISM
44	8	12	0.75	3	0.34	SYNERGISM
45	4	8	0.38	1	0.22	SYNERGISM
46	12	16	2	4	0.41	SYNERGISM
47	8	12	1	3	0.37	SYNERGISM
48	8	16	2	6	0.62	ADDITIVE
49	6	8	1	1.5	0.35	SYNERGISM
50	4	6	0.5	2	0.45	SYNERGISM
51	24	32	6	12	0.62	ADDITIVE
52	8	16	2	4	0.5	SYNERGISM
53	12	16	3	4	0.5	SYNERGISM
54	16	24	3	6	0.43	SYNERGISM
55	4	8	1	2	0.5	SYNERGISM
56	6	8	0.75	2	0.37	SYNERGISM
57	8	12	1	3	0.37	SYNERGISM
58	8	16	1	2	0.25	SYNERGISM
59	4	6	0.75	1.5	0.43	SYNERGISM
60	16	24	3	4	0.35	SYNERGISM
61	12	16	1.5	4	0.37	SYNERGISM
62	32	48	6	12	0.43	SYNERGISM
63	24	48	12	24	1	ADDITIVE
64	16	24	6	12	0.87	ADDITIVE
65	12	16	3	4	0.5	SYNERGISM
66	12	16	2	4	0.41	SYNERGISM
67	4	8	0.5	1	0.25	SYNERGISM
68	8	16	0.75	2	0.21	SYNERGISM
69	16	24	3	4	0.35	SYNERGISM
70	6	12	1.5	2	0.41	SYNERGISM
71	8	16	1.5	2	0.31	SYNERGISM
72	8	12	1	2	0.29	SYNERGISM
73	16	24	3	6	0.43	SYNERGISM
74	12	16	1.5	2	0.25	SYNERGISM
75	8	16	1	3	0.31	SYNERGISM
76	12	16	1	3	0.27	SYNERGISM
77	32	48	8	12	0.5	SYNERGISM
78	16	24	4	6	0.5	SYNERGISM
79	12	24	2	4	0.33	SYNERGISM
80	8	16	1	4	0.37	SYNERGISM
81	12	16	1.5	4	0.37	SYNERGISM
82	16	24	2	6	0.37	SYNERGISM
83	12	16	1	4	0.33	SYNERGISM
84	16	24	2	6	0.37	SYNERGISM
85	8	12	0.75	2	0.26	SYNERGISM
86	12	24	1.5	4	0.29	SYNERGISM
87	16	48	3	8	0.35	SYNERGISM
88	16	24	3	6	0.43	SYNERGISM
89	24	32	8	12	0.70	ADDITIVE
90	12	16	3	4	0.5	SYNERGISM
91	12	24	2	6	0.41	SYNERGISM
92	16	24	3	6	0.43	SYNERGISM
93	8	12	0.75	2	0.26	SYNERGISM
94	12	16	1.5	4	0.37	SYNERGISM
95	8	16	2	4	0.5	SYNERGISM
96	8	12	1.5	3	0.43	SYNERGISM
97	4	12	0.5	1.5	0.25	SYNERGISM
98	16	32	4	6	0.43	SYNERGISM
99	12	16	2	4	0.41	SYNERGISM
100	16	24	3	4	0.35	SYNERGISM

#: Isolate; FICI: FIC index; FOS: fosfomycin; MIC FOS/TEM: MIC of fosfomycin when combined with temocillin; MIC TEM/FOS: MIC of temocillin when combined with fosfomycin; TEM: temocillin. Green: synergism; yellow: additive; grey: indifferent.

**Table 2 antibiotics-13-00526-t002:** Antibiotic resistance genes identified by WGS of isolates 1, 2, 3, 4 and 10.

Strain/MLST	ARGs						Chromosomal Mutations	
	β-Lactams	Macrolide	Quinolone	Aminoglycoside	Fosfomycin	Others		
1/ST512	*bla*_KPC-3_, *bla*_SHV-11_	*mph(A)*	*oqxAB*	*aac(6′)-Ib*, *aadA2b*	*fosA5*	*sul1*, *qacE*	*acrR*	P161R, G164A, F172S, R173G, L195V, F197I, K201M (fluoroquinolones)
							*ompK36*	A217S (carbapenems) N49S, L59V, G189T, F198Y, F207Y, T222L, D223G, E232R, N304E (cephalosporins)
							*ompK37*	I70M, I128M, N230G (carbapenems)
2/ST307	*bla*_KPC-2_, *bla*_SHV-28_, *bla*_CTX-M-15_, *bla*_OXA-1_	none	*oqxAB*, *aac(6′)-Ib-cr*, *qnrB1*	*aac(6′)-Ib-cr*	*fosA5*	*sul1*, *catB3*, *dfrA14*	*acrR*	P161R, G164A, F172S, R173G, L195V, F197I, K201M (fluoroquinolones)
							*ompK36*	N49S, L59V, T184P (cephalosporins)
							*ompK37*	I70M, I128M, N230G (carbapenems)
3/ST101	*bla*_KPC-3_, *bla*_SHV-28_, *bla*_CTX-M-15_, *bla*_OXA-1_, *bla*_TEM-1B_	none	*aac(6′)-Ib-cr*, *oqxAB*, *qnrB1*	*aac(6′)-Ib-cr*, *aph(6)-Id*, *aph(3″)-Ib*, *aac(3)-IIa*	*fosA*	*catB3*, *sul2*, *tet(A)* *dfrA14*	*ompK36*	A217S, N218H (carbapenems) N49S, L59V, L191S, F207W, D224E, L228V, E232R, T254S (cephalosporins)
							*ompK37*	I70M, I128M (carbapenems)
4/ST307	*bla*_KPC-2_, *bla*_SHV-28_, *bla*_CTX-M-15_, *bla*_OXA-10_, *bla*_OXA-1_	*mph(A)*	*oqxAB*, *aac(6′)-Ib-cr*	*aac(6′)-Ib-cr*, *aadA24*, *aadA1*, *ant(2″)-Ia*	*fosA6*, *fosA5*	*sul1*, *cmlA1*, *catB3*, *ARR-2*, *dfrA12*, *dfrA14*, *qacE*	*acrR*	P161R, G164A, F172S, R173G, L195V, F197I, K201M (fluoroquinolones)
							*ompK36*	N49S, L59V, T184P (cephalosporins)
							*ompK37*	I70M, I128M, N230G (carbapenems)
10/ST unknown	*bla*_KPC-2_, *bla*_TEM-1B_, *bla*_CTX-M-14_	*mph(A)*	none	*aadA5*, *aph(6)-Id*, *aph(3″)Ib*, *aac(3)-IId*	none	*sul1*, *sul2*, *tet(A)*, *dfrA17*	*gyrA*	S83L, D87N (ciprofloxacin)
							*parC*	S80I (ciprofloxacin)

**Table 3 antibiotics-13-00526-t003:** MGEs of *K. pneumoniae* isolates 1, 2, 3, 4 and 10.

Isolate	Mobile Genetic Elements (MGEs)		
	Plasmids	Insertion Sequences (ISs)	Transposons
1	IncFIB(K) IncFII(K) IncR ColRNAI	IS5075 (IS110 family) IS102, IS903 (IS5 family) IS6100 (IS6 family) ISEc9 (IS1380 family) ISKpn1, ISSty2, ISEcI1, ISKpn21 (IS3 family) ISKpn14 (IS1 family)	Tn5403 Tn4401
2	IncFIB(K) IncC IncFII(K) IncR Col(IRGK)	IS5075 (IS110 family) IS903 (IS5 family) IS6100 (IS6 family) ISKpn1, ISEc1, ISEc15 (IS3 family) ISKpn14 (IS1 family)	Tn4401 Tn6196
3	Col(BS512) IncFIB(K) IncFII IncFII(K) IncR ColKP3 ColRNAI	IS6100, IS26 (IS6 family) ISKpn37 (IS3 family) IS30 (IS30 family) MITEEc1 (IS630 family) ISEc53, ISEc38 (ISL3 family) ISEc1, ISEc52, IS629, ISKpn8 (IS3 family) IS5, IS102 (IS5 family) IS682 (IS66 family) ISEc9 (IS1380 family)	Tn4401
4	IncFIB(K) IncC IncFII(K) IncR Col(IRGK)	IS5075 (IS110 family) IS903 (IS5 family) IS6100 (IS6 family) ISKpn1, ISEc1, ISEc15 (IS3 family) ISKpn14 (IS1 family)	Tn4401 Tn6196
10	Col(BS512) IncFIB(K) IncFII IncFII(K) IncR ColKP3 ColRNAI	IS6100, IS26 (IS6 family) ISKpn37 (IS3 family) IS30 (IS30 family) MITEEc1 (IS630 family) ISEc53, ISEc38 (ISL3 family) ISEc1, ISEc52, IS629, ISKpn8 (IS3 family) IS5, IS102 (IS5 family) IS682 (IS66 family) ISEc9 (IS1380 family)	Tn4401

## Data Availability

The original contributions presented in the study are included in the article, further inquiries can be directed to the corresponding author.

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
