# Peer review of "Synergistic Activity of Temocillin and Fosfomycin Combination against KPC-Producing Klebsiella pneumoniae Clinical Isolates"

_antibiotics, 2024, doi:10.3390/antibiotics13060526_

Round 1

Reviewer 1 Report

Comments and Suggestions for Authors

Due to the resistance of a series of KPC-producing Klebsiella pneumoniae to antibiotics, the manuscript is quite interesting. However, the authors should address some points:

1.     The authors should include the full name of KPC on line 33.

2.     On line 94, the author states that bacteria were identified by MALDI-TOF mass spectrometry; therefore, the author should incorporate the results into the manuscript.

3.     The authors should specify the number of replicates conducted in the methods section.

4.     Line 144: add “3”

5.     Line 145: 3.1

6.     Line 161: 3.2

7.     A time-kill study using combinations of both antibiotics is very interesting. The authors should select a bacterial isolate as a model and study the time-kill.

8.     In the in vivo results (graph of K. pneumoniae strain 2), there are noticeable differences in the survival rate of larvae treated with fostomycin and a combination of both antibiotics. However, these differences are not clearly discernible in the graph. The authors should provide a clearer description.

9.     In the discussion section, the author should augment the discussion by comparing it with previous publications rather than reiterating the results.

Comments on the Quality of English Language

The text with the typo can be improved.

Author Response

Due to the resistance of a series of KPC-producing Klebsiella pneumoniae to antibiotics, the manuscript is quite interesting. However, the authors should address some points:

  1. The authors should include the full name of KPC on line 33.

Reply: Ok, done

  1. On line 94, the author states that bacteria were identified by MALDI-TOF mass spectrometry; therefore, the author should incorporate the results into the manuscript.

Reply: Ok, done

  1. The authors should specify the number of replicates conducted in the methods section.

Reply: This is now stated

  1. Line 144: add “3”

Reply: Now paragraph order has changed; however, thank you for noticing

  1. Line 145: 3.1

Reply: Now paragraph order has changed; however, thank you for noticing

  1. Line 161: 3.2

Reply: Now paragraph order has changed; however, thank you for noticing

  1. A time-kill study using combinations of both antibiotics is very interesting. The authors should select a bacterial isolate as a model and study the time-kill.

Reply: The time-kill assay is a robust method to confirm synsergistic interactions, but it is very cumbersome and time-consuming. In this work we had different problems related to human and reagent resources, hence we opted for a simple, inexpensive but commonly used method for synergy testing based on MIC test strip method. In this context we would test strains that showed synergistic interactions (both in vitro and in vivo interactions) in a next study, involving other laboratory centers, for time-kill and molecular assays to elucidate the phenotypic and molecular mechanisms responsible for synergistic interactions.

  1. In the in vivoresults (graph of  pneumoniae strain 2), there are noticeable differences in the survival rate of larvae treated with fostomycin and a combination of both antibiotics. However, these differences are not clearly discernible in the graph. The authors should provide a clearer description.

Reply: We thank the reviewer for pointing this out. The description of the results has been amended to accurately describe the figure

  1. In the discussion section, the author should augment the discussion by comparing it with previous publications rather than reiterating the results.

Reply: Thank you for your comment. In the discussion paragraph, we have expanded the section where we compare our study with the two existing studies on the synergy of temocillin and fosfomycin

Comments on the Quality of English Language

The text with the typo can be improved.

Reply: Ok, done

Reviewer 2 Report

Comments and Suggestions for Authors

The authors described an alternative in vivo model to understand the activity of two antibiotics on clinical isolates.

Despite the authors declare that they did not perform the genotypic analysis I think the significance of the results obtained need to be supported by genetic analysis because the declare that they have used "100 non-repetitive" clinical isolates but it is not clear how they demonstrate this without genetic information. They assume that every single isolate expresses a different resistance mechanism but more than one couple of clinical isolates have the same the resistance profile (Table 1 Ex. 24-49, 38-48), so without any genetic information you could not consider them “different” isolates.

For all the reason I recommended the authors to give genetic information about the clinical isolates or at least the ones regarding resistance mechanisms.

I strongly suggest also the authors to explain why they decide to use strain 1, 2, 3 , 4 and 10 for the in vivo analysis because it is not clear.

Minor revisions

Lines 40, 59, 73 β-lactams instead of beta-lactams

Line 79 in vitro in italics

Line 123 K. Pneumoniae in italics

Author Response

The authors described an alternative in vivo model to understand the activity of two antibiotics on clinical isolates.

Despite the authors declare that they did not perform the genotypic analysis I think the significance of the results obtained need to be supported by genetic analysis because the declare that they have used "100 non-repetitive" clinical isolates but it is not clear how they demonstrate this without genetic information. They assume that every single isolate expresses a different resistance mechanism but more than one couple of clinical isolates have the same the resistance profile (Table 1 Ex. 24-49, 38-48), so without any genetic information you could not consider them “different” isolates.

For all the reason I recommended the authors to give genetic information about the clinical isolates or at least the ones regarding resistance mechanisms.

I strongly suggest also the authors to explain why they decide to use strain 1, 2, 3 , 4 and 10 for the in vivo analysis because it is not clear.

Reply: in the text, we stated that the 100 clinical isolates were recovered from different patients. However, it is correct that we do not have genotype analysis data, so we cannot definitively claim they are different strains. Indeed, this important limitation, correctly noted by the reviewer, has been acknowledged in the study’s limitations section.

Regarding the isolated used for the animal study: although they were not genetically tested, their antimicrobial resistance phenotype aligns with the assumption that they are different isolates. For this reason, we utilized these strains. We have now added a clarification regarding this in the text.

Unfortunately, this is the maximum we can do since it is a non-funded study.

Minor revisions

Lines 40, 59, 73 β-lactams instead of beta-lactams

Reply: Ok, done

Line 79 in vitro in italics

Reply: Usually this Journal does not want “in vitro” and “in vivo” in Italics, we will see this with the proofreading

Line 123 K. Pneumoniae in italics

Reply: Ok, done

Reviewer 3 Report

Comments and Suggestions for Authors

This manuscript appears to present some very topical information on synergism among known antibiotics. The study is simple, straight-forward, and useful. Not all readers may be familiar with the methodology used to study synergism (I was not), and the description might be improved. Also, it was not clear whether infectious agents were injected into the same proleg as the treament compounds or whether the opposite proleg was used. 

Comments on the Quality of English Language

Some minor language problems did come up. Lines 48-51 had some awkward grammar that should be written more carefully. In lines 63-64, enzymes work against antibiotics, rather than the reverse (as written here).  One or two sentences in the Discussion could use some modifications. Overall, the writing is very sound.

Author Response

This manuscript appears to present some very topical information on synergism among known antibiotics. The study is simple, straight-forward, and useful. Not all readers may be familiar with the methodology used to study synergism (I was not), and the description might be improved. Also, it was not clear whether infectious agents were injected into the same proleg as the treament compounds or whether the opposite proleg was used. 

Reply: A brief explanation of in vitro synergy tests (by MIC test strips) has been added in M&M. The top-left proleg was used for infection with K. pneumoniae, the top-right proleg was used as the injection site for antibiotics. This has been clarified.

Some minor language problems did come up. Lines 48-51 had some awkward grammar that should be written more carefully. In lines 63-64, enzymes work against antibiotics, rather than the reverse (as written here).  One or two sentences in the Discussion could use some modifications. Overall, the writing is very sound.

Reply: Thank you, we have modified the sentences in lines 48-51 and 63-64. The English in the discussion has been reviewed

Reviewer 4 Report

Comments and Suggestions for Authors

Thank you very much for the opportunity to review this manuscript entitled “synergistic activity of temocillin and fosfomycin combination against KPC-producing Klebsiella pneumoniae clinical isolates”. This study explored the synergistic effect of two antimicrobials both in vitro and in vivo. The methods were sound and the results were interesting. Besides a few typos and a few confusing sentences (Lines 126 – 127; absence of movement detected? Should it be “movement was not detected when applying stimulus”), I only have one comment. The colors in the figure were very difficult to differentiate. Could the authors please fix this.

Author Response

Thank you very much for the opportunity to review this manuscript entitled “synergistic activity of temocillin and fosfomycin combination against KPC-producing Klebsiella pneumoniae clinical isolates”. This study explored the synergistic effect of two antimicrobials both in vitro and in vivo. The methods were sound and the results were interesting. Besides a few typos and a few confusing sentences (Lines 126 – 127; absence of movement detected? Should it be “movement was not detected when applying stimulus”), I only have one comment. The colors in the figure were very difficult to differentiate. Could the authors please fix this.

Reply: Line 126 has been amended according to the reviewer's suggestion, and the figures' colors have been changed.

Round 2

Reviewer 1 Report

Comments and Suggestions for Authors

The authors have significantly improved the manuscript.

Reviewer 2 Report

Comments and Suggestions for Authors

I could understand the reason why the authors decide not to give any genetic information, but from my opinion the scientific significance of the article is not enough in this form.

I suggest the authors to consider at least the sequencing of the clinical isoltes chosen for the in vivo experiment.

Author Response

The genetic analysis has been performed as requested, demonstrating that the strains used for the animal study were different. Prof. Mariagrazia Perilli and Dr. Alessandra Piccirilli performed the genetic analysis and have therefore been added as authors.